# MSFNet: Multi-Scale Fusion Network for Brain-Controlled Speaker Extraction

## ABSTRACT

Speaker extraction aims to selectively extract the target speaker from the multi-talker environment under the guidance of auxiliary reference. Recent studies have shown that the attended speaker's information can be decoded by the auditory attention decoding from the listener's brain activity. However, how to more effectively utilize the common information about the target speaker contained in both electroencephalography (EEG) and speech is still an unresolved problem. In this paper, we propose a multi-scale fusion network (MSFNet) for brain-controlled speaker extraction, which utilizes the EEG recorded from the listener to extract the target speech. In order to make full use of the speech information, the mixed speech is encoded with multiple time scales so that the multi-scale embeddings are acquired. In addition, to effectively extract the non-Euclidean data of EEG, the graph convolutional networks are used as the EEG encoder. Finally, these multi-scale embeddings are separately fused with the EEG features. To facilitate research related to auditory attention decoding and further validate the effectiveness of the proposed method, we also construct the AVED dataset, a new EEG-Audio dataset. Experimental results on both the public Cocktail Party dataset and the newly proposed AVED dataset in this paper show that our MSFNet model significantly outperforms the state-of-the-art method in certain objective evaluation metrics.

## CCS CONCEPTS

• **Computing methodologies** → **Artificial intelligence**; • **Human-centered computing** → *Human computer interaction (HCI)*.

## KEYWORDS

Speaker extraction, EEG signals, Multi-modal fusion, Graph convolutional network, Multi-talker environment

## 1 INTRODUCTION

Sound is considered as the carrier of information. The human brain has excellent selective auditory attention capabilities, enabling individuals to extract only target auditory information while simultaneously ignoring distracting speech in a multi-speaker environment such as a cocktail party [1]. For listeners suffering from hearing loss, this presents a significant challenge. In the past decade, the rapid development of speech enhancement and speaker extraction

algorithms has propelled advancements in hearing aids [2], and as front-end speech processing techniques to extract clear attended speech for various speech applications, including voice activity detection [3], speaker diarization [4], and speech synthesis [5]. However, these methods still lack the effectiveness of human selective attention neural mechanisms and may be constrained by environmental limitations in practical applications.

In recent years, remarkable progress has been made in the field of speech separation [6]. It is designed to separate the voice of a single speaker from scenarios where multiple speakers are talking simultaneously, such as Conv-tasNet [7], Sepformer [8], and TF-GridNet [9]. Most speech separation algorithms require prior information of the number of speakers in the mixture and consider label permutation problem, greatly limiting the practicality of these methods. Furthermore, the separated speech source is independent of the listener's attention selection. Speech separation networks separate all sound sources but cannot determine which speaker is the target and which are interference. However, in certain acoustic scenarios, listeners only pay attention to one speaker. This necessitates a subsequent speaker verification system to utilize given target speaker information, including neural signals [10] or visual attention [11], for speech tracking, thereby further increasing computational complexity.

The speaker extraction adopts a distinct strategy by employing a speaker encoder that emulates the top-down intentional focus, augmenting the acoustic signal with additional informative signals. It selectively extracts the speech of the target speaker based on the provided reference cues, thereby avoiding the aforementioned issues. Common auxiliary reference cues include pre-registered unseen target speech [12], observable lip movements [13], spatial location information [14], or understanding of contextual relevance [15]. However, these cues cannot automatically separate the desired speaker based on individual subjective awareness. The use of target speaker utterances is constrained by the necessity of prior information about the identity of target speakers in the scene, and listeners are also not able to keep their eyes fixed on the speaker they are interested in.

In order to extract the target speech from the mixture speech of multiple speakers without any registered prior information such as the identity of target speaker, a proposed solution is to decode the brain neural signals of listeners to determine the target speaker, thereby endowing the system with active perceptual abilities. According to the latest research in neuroscience, it proves that auditory attention of listeners can be decoded from recorded brain activity [16]. EEG provides a non-invasive and effective method for studying cortical neural activity, making it particularly suitable for auditory attention detection (AAD) tasks [17]. Earlier studies [18] were predominantly dedicated to enhancing the performance of the cascaded approach involving blind source separation and AAD. Estimating the speech envelope of the target speaker from the listener's

*ACM MM, 2024, Melbourne, Australia*
© 2024 Copyright held by the owner/author(s). Publication rights licensed to ACM.
ACM ISBN 978-x-xxxx-xxxx-x/YY/MM
https://doi.org/10.1145/nnnnnnn.nnnnnnn

EEG signals, the estimated envelope is then compared individually with each separated source to identify the most closely matched speaker. However, the performance of this method is heavily reliant on the accuracy of AAD.

In this paper, we introduce a multi-scale fusion network (MSFNet) for brain-controlled speaker extractionan, a end-to-end time-domain model. The MSFNet method models the attention direction of listeners directly through the recorded EEG signals to extract target speech. It consists of four components: speech encoder, EEG encoder, speaker extraction network, and speech decoder. To fully leverage speech information and more accurately capture the temporal characteristics of speech, the speech encoder encodes a segment of mixed speech waveform into multi-scale speech embeddings with different time scales. In the EEG encoder, graph convolutional networks (GCN) are used to effectively extract the non-Euclidean data from EEG trials, obtaining a feature representation of target speaker information. Finally, in the speaker extraction network, these multi-scale speech embeddings are separately fused with the EEG features and estimates corresponding receptive masks for extracting the target speaker. Experimental results on the main Cocktail Party dataset show that the proposed MSFNet model achieves 11.5% and 13.6% relative improvements over the state-of-the-art method in terms of scale-invariant signal-to-distortion ratio (SI-SDR) and perceptual evaluation of speech quality (PESQ).

The main contributions of this paper can be summarized as follows:

- We propose a multi-scale fusion network for brain-controlled speaker extraction, where speech features with different time scales are separately fused with EEG features to extract the target speaker, thereby enhancing the perception and quality of the speech.
- We propose a new Audio-Video EEG dataset, referred to as AVED dataset, to facilitate research in auditory attention decoding and brain-controlled speaker extraction. To simulate real-world perceptual environments, the AVED dataset includes scenarios where both video and audio stimuli are provided, as well as scenarios where only audio stimuli are provided, offering richer modal information.
- Experimental results on the public Cocktail Party dataset and our proposed AVED dataset both demonstrate that our MSFNet model achieves significant improvements over the baseline methods.

## 2 RELATED WORK

This section primarily reviews the background of the speaker extraction task, briefly summarize the development of brain-controlled speaker extraction and outline various current approaches.

### 2.1 Speaker Extraction

Emulating the human auditory system, speaker extraction technology incorporates an additional auxiliary network designed to extract voiceprint embedding vectors, including i-vector [19], x-vector [20], and d-vector [21], with distinct speaker identity characteristics. When given an arbitrary-length speech segment, the speaker encoder learns the speaker embedding of the speaker of interest to the listeners and selects the corresponding speech signal

from a multi-talker speech. For example, in VoiceFilter [22] and TseNet [23], a pre-trained speaker encoder is constructed to extract d-vector and i-vector as features representing the target speaker. But the pre-trained speaker encoder operates independently from the overall speaker extraction network, leading to a loss of crucial attended speaker information. Then in SpEx [24], the different idea of the context of speaker extraction is introduced, involving joint training and optimization of the speaker encoder and speaker extraction network within a multi-task learning framework.

Visual information can also serve as a reference clue for the target source, immune to the disruption caused by acoustic noise and speech interference. In accordance with the actual scenario, it is reasonable to assume that if there is a facial image, the person visible in the image is likely the one being attended to. The Conversation [25] and Time-domain Speaker Extraction Network (TDSE) [26] are examples of audio-visual speech extraction that pretrain a visual encoder to encode the lip image sequence of the target speaker into visemes, ensuring temporal synchronization between lip movements and speech. However, the challenge of extracting the target speech from the multi-speaker environment without prior information about speakers identity remains unresolved. Moreover, for practical applications such as hearing aids, there is a lack of direct connection with the human brain, preventing the detection of attention-related information based on individual subjective consciousness. To assist listeners with hearing impairments and to promote the development of neuro-steered hearing aids devices, researching how speaker extraction algorithms utilize EEG signals as input for brain-computer interaction would be highly meaningful.

### 2.2 Brain-Controlled Speaker Extraction

Research in cognitive neuroscience suggests a mapping relationship between attended speech stimuli and brain-evoked neural data. In some approaches [27][28][29], taking advantage of the fact that the target speaker can be decoded from brain activity, the speech envelope of interested speaker is first estimated from listener's EEG signals, and then the estimated speech envelope is compared with each separated sources, determining the one with the highest similarity as the target speech and subsequently amplifying it. These methods still do not circumvent the drawback of requiring knowledge of the number of speakers in blind source separation, adding unnecessary computational complexity as an additional concern.

For the optimization of different problems, the brain-controlled speaker extraction method is constantly refined. The brain-inspired speech separation (BISS) [30] model designs a brain decoder as a guiding network to extract the speech envelope of the target speaker from EEG recordings. Taking the speech mixture and the decoded neural envelope as inputs, the target extraction network jointly performs the speaker selection process of AAD and speaker separation in the frequency domain. This may cause potential phase errors. The approach of training the two networks separately also imposes limitations on the algorithm's performance. The brain-enhanced speech denoiser (BESD) [31] model and U-shaped BESD (UBESD) [32] model mainly investigate the problem of brain-driven speech enhancement in multi-speaker environments, entirely executed in the time domain, exhibiting superior performance. The use of feature-wise linear modulation (FiLM) [33] between EEG signals

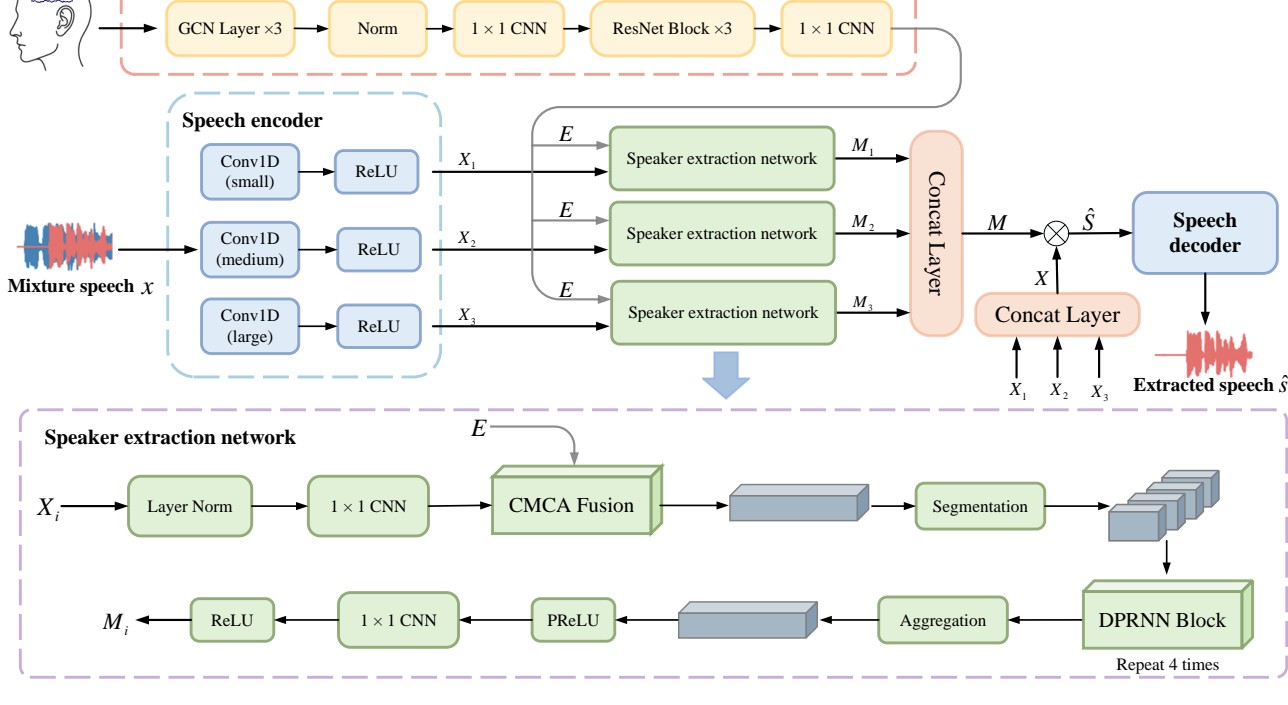

**Figure 1: The overall block diagram of the proposed MSFNet model. It consists of a speech encoder, an EEG encoder, a speaker extraction network, and a speech decoder. The symbol ⊗ represents element-wise multiplication of features.**

and the mixture dynamically adjusts feature mappings in the neural network, aiding in extracting more comprehensive features of the attended speaker. Currently, the brain-assisted speech enhancement network (BASEN) [34] utilizes the Conv-TasNet backbone to construct both the separation network and the EEG encoder, and it proposes a convolutional multi-layer cross attention (CMCA) module intricately fuse EEG with speech features, strengthening the correlation between them.

## 3 THE PROPOSED MSFNET

### 3.1 Problem Formulation

Given a multi-speaker mixed speech signal $x$, which includes the target speech signal $s$ and interfering speech signals $b_i$:

$$x = s + \sum_{i=1}^{I} b_i \in \mathbb{R}^{T_s} \tag{1}$$

where $I$ represents the number of interfering speakers in the scenario, while $T_s$ denotes the time length of mixture speech segments. In this work, we do not consider the additional influence of background noise or room reverberation. In addition, using the EEG signal $e \in \mathbb{R}^{N*T_r}$ as an auxiliary reference cue, $N$ is the number of EEG channels, and $T_r$ is the time length of the EEG signal. EEG data and speech stimuli are recorded synchronously for the same duration, ensuring temporal alignment between the two. The difference between $T_s$ and $T_r$ reflects the disparity in sampling rates

between EEG and audio signals, which will be taken into account during the data preprocessing stage.

By fusing the mixed speech signal $x$ with attentional information about the target speaker from the EEG signal $e$, enabling multimodal training for complementary learning, the final goal of brain-controlled speaker extraction is to reconstruct $\hat{s}$ as closely as possible to the true source $s$ in $x$.

### 3.2 Overall Architecture

In Figure 1, we propose the MSFNet model, a new brain-controlled speaker extraction network with audio-EEG fusion at multiple scales, which includes speech encoder, EEG encoder, speaker extraction network, and speech decoder.

Drawing inspiration from the concept of multi-modal speaker extraction, we follow an end-to-end network of encoder-decoder to comprehensively extract fused features from both EEG and speech, filtering out the target speech based on listener's attention information. In this paper, we design a new brain-controlled speaker extraction system named MSFNet network, which includes a multiple-branch network structure for extracting multi-scale speech feature representations. In each branch, the same EEG features are separately incorporated for fusion to obtain combined information about speech at different time scales and EEG signals, further enhancing the quality of the target speech. We adopt a network design based on the time-domain, which avoids the phase estimation issues associated with frequency-domain methods.

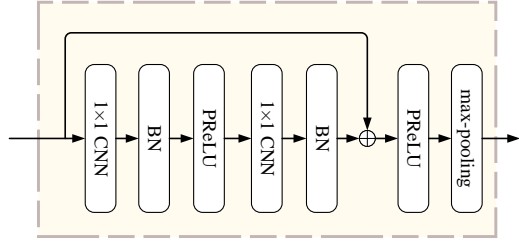

**Figure 2: The components of the ResNet Block in the EEG encoder. The symbol ⊕ denotes element-wise addition.**

Here, the introduction of each component is as follows: 1) The multi-scale speech encoder transforms mixture speech samples $x$ into speech embeddings $X_1$, $X_2$, and $X_3$ with different time resolutions. 2) The EEG encoder encodes the N-channel EEG signal $e$ into a low-dimensional feature representation $E$, referred to as EEG embedding, providing attentional selection capability for the speaker extraction network. 3) For the three outputs $(X_1, X_2, X_3)$ of the multi-scale speech encoder, the network is designed with three branching paths. In each speaker extraction network, dual-modal features are first fused, followed by the individual estimation of a corresponding mask $M_i$ ($i = 1, 2, 3$) that only allows the target speech to pass in $X_i$ ($i = 1, 2, 3$). Finally, $M_1$, $M_2$, and $M_3$ are concatenated to obtain a multi-scale mask $M$. 4) The masked speech embedding $\hat{S}$ are converted by the speech decoder into time-domain speech waveforms $\hat{s}$.

*3.2.1  Speech Encoder.* The speech encoder consists of three 1-dimensional convolutional layers (conv1D) with different filter lengths, which can learn the feature representation at various scales. In most cases, the most crucial information is contained in the frequency of a signal. However, during the process of using the short-time Fourier transform (STFT), a trade-off must be made between temporal resolution and frequency resolution, requiring a careful balance based on specific needs. Especially for time-varying non-stationary signals, small windows are suitable for high frequencies, while large windows are suitable for low frequencies. Adopting a multi-resolution speech representation method allows for comprehensive coverage of the time-frequency information in speech and more accurately captures the temporal characteristics of speech.

By using a couple of parallel conv1D with N filters, each dedicated to a different temporal resolution, followed by a rectified linear unit (ReLU) activation function, the input mixed speech $x \in \mathbb{R}^{T_s}$ can be encoded into three speech embeddings $X_i$, which can be defined as:

$$X_i = ReLU(conv1D(x, 1, N, L_i)) \in \mathbb{R}^{T_t}, i = 1, 2, 3 \qquad (2)$$

where conv1D has input channels 1, output channels $N$, kernel size $L_i$. To fully capture features at multiple scales, we perform concatenation operations on speech embeddings in the later part of the network. Therefore, it is essential to ensure the same $L_1/2$ stride in parallel convolutions. A window of speech segments containing $L_i$ samples will shift by $L_1/2$ samples each time.

In this paper, only three different time scales are studied. The three branches of the network use convolutional kernels of three sizes: $L_1$-small, $L_2$-medium, and $L_3$-large, covering various window lengths. This configuration shows good generality.

*3.2.2  EEG Encoder.* The EEG encoder is designed to learn EEG embedding $E$ from the input EEG signal $e$ that exhibit temporal correlations with the interested speech. We do not explicitly reconstruct the target speech envelope but instead extract feature representations from the EEG signals that capture aspects relevant to attentive listening in the brain.

In more detail, the first part of the EEG encoder comprises three GCN layers. Modeling multi-channel EEG features using a graph, each electrode in the EEG data is considered as a node. The method dynamically learns intrinsic relationships between different EEG channels, representing them with an $N \times N$ adjacency matrix to extract more distinctive EEG signal features. Then, the adjacency matrix, which captures distinctive features, is utilized to improve the accuracy of attentional information for the target speaker.

In the EEG encoder, immediately following is a $1 \times 1$ CNN and a set of three residual network (ResNet) blocks. The ResNet block is shown in Figure 2, which consists of two $1 \times 1$ CNNs, a batch normalization (BN) layer, parametric ReLU (PReLU) and a max-pooling layer with kernel size 1×3. After passing through the second BN layer, the intermediate activation features must be further added to the input of the first conv1D through a skip connection. At each conv1D layer, normalizing the output is necessary to address the issues of gradient vanishing and exploding. PReLU activation is applied to the sum of the skip connection and the output of the second convolutional layer. The use of a max-pooling layer has reduced the temporal dimension of the input features by a factor of 3, focusing on capturing the most crucial features. Finally, a $1 \times 1$ CNN is employed to map the features to a fixed dimension, obtaining the downsampled EEG embedding $E$.

The entire process can be defined as follows:

$$E = EEGencoder(e) \in \mathbb{R}^{C \times T_p} \qquad (3)$$

*3.2.3  Speaker Extraction Network.* The speaker extraction network consists of three independent branching paths. It is worth noting that the parameters of the speaker extraction network used for processing speech features at three time scales are shared, so this helps to control the model's parameter number to some extent. In each path, the fusion of speech features with distinct time scales and EEG features is performed first. Subsequently, the fused embeddings are fed into the dual-path recurrent neural network(DPRNN)[35] to estimate target speaker masks $M_i$ at different time scales, selectively allowing the target speech to pass through in the multi-scale speech embeddings $X_i$. To further integrate attended speech information across multiple time resolutions, it is necessary to concatenate the three masks estimated by the speaker extraction network. This results in an intermediate tensor of size $3N \times T_t$. Following this, channel-wise normalization and a conv1D layer is utilized to obtain the final mask $M$, which is used to filter out interfering speakers. The same operation is applied to the three speech embeddings $X_i$ learned by the speech encoder simultaneously, resulting in the multi-scale speech embedding $X$. The masked speech embedding $\hat{S}$ is calculated through element-wise multiplication between $X$ and

$M$:

$$\hat{S} = X \otimes M \in \mathbb{R}^{N \times T_t} \quad (4)$$

where

$$X = concat(X_1, X_2, X_3), M = concat(M_1, M_2, M_3) \quad (5)$$

For the fusion of dual-modal features, we use the CMCA-based cross attention mechanism proposed in BASEN. Between two adjacent layers, there are multiple pairs of cross-attention blocks with skip connections and group normalization. In CMCA, the left and right branches handle the audio and EEG streams, respectively. The features from both branches are added layer by layer. In the end, the two features obtained through layer-wise addition, along with the original audio embedding and the original EEG embedding, are concatenated along the channel dimension to construct the fused features.

Temporal convolutional network (TCN) and DPRNN are highly popular speech separation networks. Considering the limitation of one-dimensional convolution with a fixed receptive field, which becomes inadequate for modeling longer global dependencies when the receptive field is smaller than the length of the speech sequence, we employ the DPRNN model in the mask estimation process to efficiently capture long-sequence speech patterns. DPRNN optimizes RNNs in deep models by segmenting long input sequences into smaller chunks and iteratively applying intra-chunk modeling and inter-chunk modeling. The input length is proportional to the square root of the original sequence length in each operation. The speaker extraction process mainly consists of three stages: the segmentation of input speech embeddings, internal operations within 4 repeated DPRNN blocks, and the overlapping summation of sequence segments.

*3.2.4 Speech Decoder.* The speech decoder reconstructs time-domain speech samples $\hat{s}$ from masked speech embeddings . In the speech encoder, padding operations are applied to ensure uniform dimensions for the three speech embeddings after convolutional layers. Consequently, in the decoder, there is no need to address the problem of using filters with varying lengths in the encoder. It primarily entails a deconvolution, in contrast to the encoder, with $N$ input channels and 1 output channel, a kernel size of $L_1$, and a stride size of $L_1/2$:

$$\hat{s} = deconv1D((\hat{S}, N, 1, L_1), stride = L_1/2) \in \mathbb{R}^{1 \times T_s} \quad (6)$$

## 3.3 Objective Loss Function

We employ a SI-SDR[36] loss as a measure of the error between the reconstructed speech and the ground-truth. It has been demonstrated to perform well in time-domain speech separation algorithms and is widely used. SI-SDR can be formulated as follows:

$$SI\text{-}SDR = 10log_{10} \frac{\|x_{target}\|^2}{\|x_{res}\|^2} \quad (7)$$

where

$$x_{target} = \frac{\hat{s}^T s}{\|s\|^2} s \quad (8)$$

$$x_{res} = x_{target} - \hat{s} \quad (9)$$

in which $\hat{s}$ and $s$ stand for the extracted and true signals of the target speaker, respectively. To ensure the stability of model training,

**Table 1: The summary of settings details for each dataset.**

| Cocktail Party Dataset | |
| --- | --- |
| Number of subjects | 33 |
| Subjects gender | 28 males and 5 females |
| Speakers gender | males |
| EEG channels | 128 |
| EEG trials of each subject | 30 |
| Duration of each EEG trial | 60s |
| Language of stimuli | English |
| AVED Dataset | |
| Number of subjects | 20 |
| Subjects gender | 14 males and 6 females |
| Speakers gender | male and female |
| EEG channels | 32 |
| EEG trials of each subject | 16 |
| Duration of each EEG trial | 152s |
| Language of stimuli | Mandarin |

before calculating SI-SDR, the signals $\hat{s}$ and $s$ need to be normalized to zero mean. This is because the scale of the target speech may change after processing. In general, a higher SI-SDR value indicates better quality of the reconstructed speech. We aim to minimize the negative SI-SDR as the training objective.

# 4 EXPERIMENTAL SETUP

## 4.1 Datasets

*4.1.1 Cocktail Party Dataset.* The first dataset used in this experiment is obtained from the authors of [37]. The data collection procedures are performed in accordance with the Declaration of Helsinki and are approved by the Ethics Committees of Trinity College Dublin. A total of 33 subjects (28 males and 5 females) with normal hearing and no history of neurological disorders take part in the experiment. The average age is $27.3 \pm 3.2$ years. The EEG data of the sixth subject is excluded due to recording noise interference. Each subject undertakes 30 trials, with each trial lasting 60 seconds. During the collection process, subjects simultaneously listen to two different classic stories, one presented to the left ear and the other to the right ear. Each story is narrated by a different male speaker. The subjects are evenly divided into two groups, with one group instructed to focus on the left ear (17 people) and the other on the right ear (16 + 1 excluded subject).

All speech stimuli are presented monophonically at a sampling rate of 44.1 kHz. The EEG data, collected from 128 channels, are originally recorded at a sampling rate of 512 Hz and later reduced to 128 Hz. For each subject, 5 trials are randomly chosen for the test set, 2 trials for validation, with the remaining data serving as the training set.

*4.1.2 AVED: Audio-Video EEG Dataset.* We propose the AVED dataset, a new EEG-Audio dataset designed for tasks related to auditory attention decoding, with all subjects signing informed consent forms. On this dataset, we further validate the effectiveness of the proposed method. Subsequently, we also ensure the

**Table 2: The comparison of various methods on the Cocktail Party dataset and AVED dataset. BASEN\* is our re-implementation, and UBESD on the AVED dataset is also our re-implementation.**

| Methods | Cocktail Party | | | | | AVED | | | | |
|---|---|---|---|---|---|---|---|---|---|---|
| | SI-SDR(dB) | SDR(dB) | STOI | ESTOI | PESQ | SI-SDR(dB) | SDR(dB) | STOI | ESTOI | PESQ |
| Mixture | 0.45 | 0.47 | 0.71 | 0.55 | 1.61 | 1.72 | 1.73 | 0.76 | 0.63 | 1.58 |
| BESD[31] | 5.75 | _ | 0.79 | _ | 1.79 | _ | _ | _ | _ | _ |
| UBESD[32] | 8.54 | _ | 0.83 | _ | 1.97 | 7.89 | 8.10 | 0.85 | 0.72 | 1.75 |
| BASEN*[34] | 11.56 | 11.66 | 0.86 | 0.72 | 2.21 | 8.46 | 8.68 | 0.86 | 0.75 | 1.91 |
| MSFNet(ours) | **12.89** | **13.03** | **0.88** | **0.77** | **2.51** | **9.65** | **9.84** | **0.89** | **0.79** | **2.07** |

accessibility and availability of this dataset to foster collaboration, reproducibility and further advancements in the field. A total of 20 normal hearing subjects (14 males and 6 females) take part in the data collection, with a mean age of 20 years. Each subject undertakes 16 trials, with each trial lasting 152 seconds (including 2s silence at the beginning of the experiment). All auditory stimuli are derived from 16 stories selected from a collection of Chinese short stories. In each trial, audio recordings of two different stories, one read by a male and the other by a female, are simultaneously presented to the subjects. Subjects determine the attentional direction based on instructions and ensure that the story they focus on did not overlap. After each trial, three multiple-choice questions related to the story the subject focused on are provided to ensure their attention during the experimental process.

The audio signals are uniformly set to a sampling rate of 44.1kHz and presented at the same volume level. We downsample the speech stimuli from both the left and right ears to 14.7 kHz, and then combine them to simulate a speech mixture. 32-channel EEG data are recorded at a rate of 1kHz and further downsampled to 128 Hz. For each subject, the trials focusing on the male talker are selected for experimental evaluation to avoid instability caused by attention switches. Then, we divide all trials of each subject into training, validation, and test sets with proportions of 75%, 12.5%, and 12.5%, respectively.

In Table 1, a summary of the settings details for each dataset is provided.

## 4.2 EEG Preprocessing

For the Cocktail Party dataset, our preprocessing steps remain consistent with UBESD. First, The original EEG data are band-pass filtered from 0.1Hz to 45Hz to retain only the relevant frequency bands. Identifying channels with excessive noise, we recalculate those channels using spline interpolation based on the surrounding channels. The EEG data are re-referenced using the average of mastoid channels to avoid introducing noise and loss of information. In EEGLAB [38], we perform independent component analysis (ICA) to remove artifacts from eye movements and muscle activity. Among them, trial data containing excessive noise interference are being excluded from the experiment for each subject. Additionally, considering potential sources of interference in EEG signals that may not directly correlate with speech stimuli, it is preferable to record activity that includes information more relevant to the target speech stimuli, rather than directly using the raw EEG signals

as input for the proposed model. Therefore, we further employ a frequency-band coupling model [39] to extract potential brain neural activities more associated with the sound stimuli from the EEG data, represented as cortical multiunit neural activity (MUA).

For the AVED dataset, we initially apply a notch filter to the original EEG data to eliminate power frequency interference at 50 Hz. We use FIR filters for high-pass and low-pass filtering to remove voltage drift and high-frequency noise. This series of filtering steps contributes to extracting clean EEG signals. The ICA method is similarly performed to remove potential mixed components. Finally, the EEG signals from all channels are re-referenced using an average reference to eliminate the error impact caused by changes in the original reference electrodes.

## 4.3 Training Details

We implement the network configuration and conduct experiments using the PyTorch framework on two NVIDIA GeForce RTX 3090 GPUs. All models are trained for 60 epochs with a batch size of 8. We use Adam optimizer with a maximum learning rate of 0.00035 and a weight decay of 0.001. The learning rate adjustment strategy involves linear warm-up followed by cosine annealing, where the warm-up ratio is 4%. The periodicity of cosine annealing is determined by the total epochs and the divider, with a divider value set to 25 in this experiment. To adapt to different datasets, we make some modifications to the experimental parameters. On the AVED dataset, we adjust the maximum learning rate for model training to 0.001 and set the weight decay to 0.01.

For the MSFNet network, the values of $N$ and $C$ are set as 128 and 64. In the speech encoder, the three conv1D layers have different kernel sizes, namely $L_1 = 0.0025s$, $L_2 = 0.01s$, and $L_3 = 0.02s$. These small, medium, and large windows cover 36, 147, and 294 samples, respectively. For the training and validation sets, the data from each trial is sliced into 2s segments. During inference, speech stimuli and EEG data are divided into 20s segments. Without any overlap of data between sets.

## 4.4 Evaluation Metrics

We mainly use three objective evaluation metrics to measure the quality of the extracted speech in our proposed method, including SI-SDR, PESQ [40], and short term objective intelligibility (STOI) [41]. Among these, SI-SDR is the main metric as it is highly suitable for single-channel speech separation or enhancement tasks and

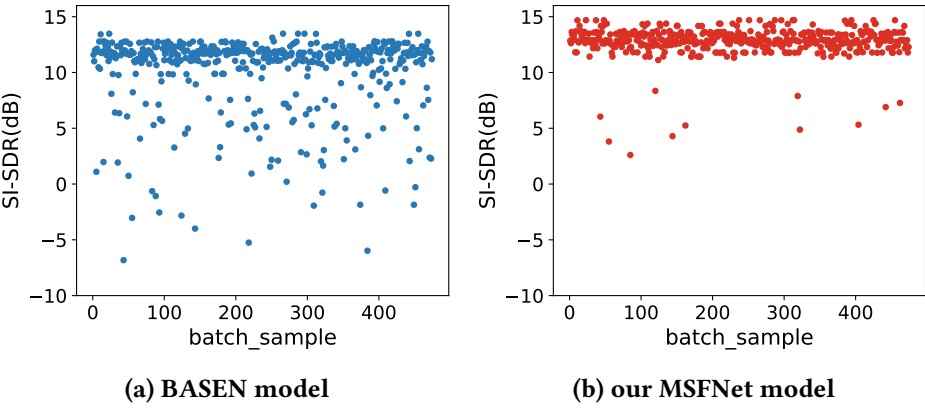

(a) BASEN model

(b) our MSFNet model

Figure 3: Visualization of experimental results on the Cocktail Party dataset, by (a) the baseline BASEN model, represented in blue color; (b) our proposed MSFNet model, represented in red color.

possesses good robustness. PESQ and STOI can predict the human-perceived quality and intelligibility of the speech, respectively. Additionally, in order to comprehensively validate the efficacy of our approach, we also use the signal-to-distortion ratio (SDR) and extended short-time objective intelligibility (ESTOI) metrics in our comparative experiments with the baselines and some ablation study. For all these objective metrics, a higher value indicates better algorithm performance.

## 5 EXPERIMENTAL RESULTS

To validate the effectiveness of the proposed method, we conduct several sets of experiments and mainly evaluate the SI-SDR, STOI, and PESQ metrics. It can be summarized as: comparison of our method with baselines, difference comparison between speaker extraction module using DPRNN structure and TCN structure, and ablation experiment. It is worth noting that we only conduct comparative experiments with the baseline on the AVED dataset; all other experiments in this work are performed on the main Cocktail Party dataset.

### 5.1 Comparison with Baseline Models

*5.1.1 Experimental Results on Cocktail Party Dataset.* The experiments conducted on this dataset utilize data from all subjects, and the network training process does not provide any prior information about the identity of the target speaker, achieving a subject-independent configuration, i.e., unknown attended speaker extracting. We compare the performance of the proposed method with other baseline methods, namely three major time-domain approaches: BESD, UBESD, and BASEN. The results are drawn in Table 2. The results indicate that our proposed MSFNet model has a relative improvement of 1.33 dB, 0.02, and 0.3 in SI-SDR, STOI, and PESQ, respectively, compared to the state-of-the-art BASEN method.

In Figure 3, we further depict scatter plots of the SI-SDR values for the extracted attended speech from both the proposed model and the BASEN model. It can be readily observed that, for the BASEN method, the average SI-SDR values of all samples are slightly lower

compared to those of the MSFNet method. Additionally, some samples even yield negative SI-SDR values, indicating that this model makes some speaker confusion errors during the speaker extraction process, consequently affecting the model performance. In contrast, for our proposed method, all samples have positive SI-SDR values and are concentrated in distribution. This implies that the proposed MSFNet model consistently extracts the correct target speaker and ensures high signal quality. It can be demonstrated that the MSFNet network can more comprehensively fuse mixed audio and attentional information about the target speaker learned from various channels of EEG signals.

*5.1.2 Experimental Results on AVED Dataset.* As we only use trial data where subjects pay attention to the same speaker in the AVED dataset for network training and testing inference, we refer to this experimental setup as speaker-dependent extraction, i.e., known attended speaker extracting. In this setup, we similarly compare the proposed method with UBESD and BASEN, and the results are also presented in Table 2. The results on the AVED dataset also demonstrate that our proposed MSFNet model achieves relative improvements of 1.19 dB, 0.03, and 0.16 in terms of SI-SDR, STOI, and PESQ compared to the BASEN method, respectively. Therefore, we can conclude that, across different datasets and experimental setups, the MSFNet model continues to exhibit competitive performance compared to other existing audio-EEG multimodal speaker extraction baselines.

### 5.2 The Impact of Different Speaker Extraction Networks

In the speaker extraction network that using EEG signals and speech for multi-modal fusion, we compare the performance of using DPRNN architecture and TCN architecture. The MSFNet network utilizes four repeated DPRNN blocks for estimating the receptive masks, while employing a stack of TCN blocks, based on Depth-wise conv1D layer, repeated four times, to estimate the receptive mask, forming the network called MSFNet (TCN). As shown in Table 4, we can clearly observe that the proposed MSFNet significantly outperforms MSFNet (TCN) across all metrics. Here,

**Table 3: Ablation study between single-scale and multi-scale. $L_1$, $L_2$ and $L_3$ respectively represent the sample counts included in the different filter lengths of the convolution in the speech encoder.**

| Single vs Multiple Scales | L1 | L2 | L3 | SI-SDR(dB) | SDR(dB) | STOI | ESTOI | PESQ |
|---|---|---|---|---|---|---|---|---|
| Single Scale | 36 | _ | _ | 12.21 | 12.37 | **0.88** | 0.74 | 2.34 |
| Single Scale | _ | 147 | _ | 12.15 | 12.30 | 0.87 | 0.73 | 2.33 |
| Single Scale | _ | _ | 294 | 12.17 | 12.32 | 0.87 | 0.74 | 2.33 |
| Dual Scales | 36 | 147 | _ | 12.58 | 12.72 | 0.87 | 0.75 | 2.47 |
| Dual Scales | 36 | _ | 294 | 12.42 | 12.54 | **0.88** | 0.76 | 2.41 |
| Dual Scales | _ | 147 | 294 | 12.58 | 12.75 | **0.88** | 0.73 | 2.38 |
| Multiple Scales | 36 | 147 | 294 | **12.89** | **13.03** | **0.88** | **0.77** | **2.51** |

**Table 4: Comparative study of DPRNN and TCN in speaker extraction networks on the Cocktail Party dataset. All metrics are higher the better.**

| Methods | Speaker extraction | SI-SDR(dB) | SDR(dB) | STOI | ESTOI | PESQ |
|---|---|---|---|---|---|---|
| Mixture | _ | 0.45 | 0.47 | 0.71 | 0.55 | 1.61 |
| MSFNet(TCN) | TCN | 10.27 | 10.53 | 0.82 | 0.70 | 2.03 |
| MSFNet(ours) | DPRNN | **12.89** | **13.03** | **0.88** | **0.77** | **2.51** |

to maintain consistency in the separation network, we only replace the corresponding DPRNN blocks with TCN blocks stacked four times, each with the same recursive depth as in the DPRNN blocks. This may lead to the lower performance of MSFNet (TCN) using the TCN structure in Table 4 compared to the BASEN model using the same TCN structure.

## 5.3 Ablation Study

*5.3.1 Effect of the GCN Layer in EEG Encoder.* To illustrate that incorporating GCN layers into the EEG encoder contributes to learning the correlations between different brain regions and enhances speaker extraction performance, we compared the methods with or without GCN in Table 5. We can see that the network without using GCN achieves only 11.89 dB of SI-SDR, which is 0.11 dB lower than when using only one layer of GCN. To fine-tune the optimal number of GCN layers, we also assessed the impact of layer numbers ranging from 1 to 4 on the experimental results. When using 3-layer GCN, the model achieves the best performance across all metrics.

*5.3.2 Single-Scale versus Multi-Scale.* Next, we will explore the effectiveness of the idea of separately fusing multi-scale speech embeddings and EEG embedding. In Table 3, we can observe that the combination of filters covering three different time-frequency resolutions performs the best, with an SI-SDR of 12.89dB, STOI of 0.88, and PESQ of 2.51. Furthermore, in the comparison of experimental results under the single-scale speech encoder setting, it is indicated that using filters with a length of 36 samples (about 0.0025s) to implement a small window yields the best system performance. The respective values for SI-SDR, STOI, and PESQ are 12.21dB, 0.88, and 2.34. With an increase in the number of filters, such as jointly using filters with a length of 36 samples and filters with a length of 147 samples (0.01s), the experiments achieved even better results. This

**Table 5: Ablation study on different GCN layer numbers in the EEG encoder.**

| GCN layers | SI-SDR(dB) | STOI | PESQ |
|---|---|---|---|
| 0(without) | 11.89 | 0.86 | 2.32 |
| 1 | 12.00 | 0.87 | 2.28 |
| 2 | 12.58 | 0.87 | 2.42 |
| 3 | **12.89** | **0.88** | **2.51** |
| 4 | 12.64 | **0.88** | 2.39 |

finding is similar to the speaker extraction task [24] and speech recognition experiments [42]. Therefore, to benefit from different time-frequency information, we adopt a multi-branch structure in the network. In each branch, the speech encoder utilizes CNNs with different filter lengths to represent small, medium, and large scales.

## 6 CONCLUSION AND FUTURE WORK

In this paper, we propose a network that fuses audio and EEG signals at multiple scales to effectively cover the time-frequency resolutions of target speech. To better understand and investigate the selective auditory attention mechanism in the human brain, we propose a new EEG dataset for evaluating and improving brain-controlled speech extraction techniques. We also propose the use of GCN in the EEG encoder to endow the model with a certain level of spatial understanding of the structure of EEG trials. Experiments conducted on two datasets demonstrate the improvement of our MSFNet method compared to existing techniques. In the future, we plan to explore more effective fusion methods for speech and EEG signals to enhance the accuracy of target speaker extraction and improve the quality of the extracted speech.

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
