# OpenReview forum: "MSFNet: Multi-Scale Fusion Network for Brain-Controlled Speaker Extraction"
_acmmm.org/ACMMM/2024/Conference — MM2024 Poster_

### Official Review · Reviewer_LmY4 · 2024-05-21

**Rating:** 4
**Confidence:** 3

**Summary:**

This paper presents an approach to target speaker extraction using EEG signal based on selective auditory attention. They propose a multi-scale fusion network (MSFNet) to effectively fuse audio and EEG signals.

**Strengths:**

The paper is well-written, with a clear structure and easily understandable language.
The integration of EEG signals for target speaker extraction is indeed a promising direction.

**Limitations:**

1. There are already some EEG-based speaker extraction works.
[1] NeuroHeed: Neuro-steered speaker extraction using eeg signals (ArXiv)
[2] NeuroHeed+: Improving neuro-steered speaker extraction with joint auditory attention detection (Published in ICASSP2024)
I think it's not proper to claim that EEG-based target speaker extraction is still an unresolved problem. It's not insight and novel enough to be the motivation.

2. The manuscript would benefit from a more comprehensive literature review.
The brain-informed speech separation (BISS) [3] is the pioneering work that directly extracts the attended speech signal based on the EEG signal.
[3] Brain-informed speech separation (BISS) for enhancement of target speaker in multi-talker speech perception

I recommend that the authors revise their motivation and novelty claim, providing a clearer differentiation from prior work and a more detailed comparison with key methodologies like BISS.

**Suitability:**

2

---

### Official Review · Reviewer_RpuR · 2024-05-23

**Rating:** 4
**Confidence:** 4

**Summary:**

An MSFNet is proposed to achieve the multi-scale fusion of the EEG to extract the speeches for certain speakers.

**Strengths:**

The task of speaker extraction through electroencephalogram (EEG) signals is indeed fascinating and presents a novel approach to the field of speaker extraction.

**Limitations:**

Only EEG-related approaches are considered as baselines, more recent works are required.

**Suitability:**

2

---

### Official Review · Reviewer_gJFY · 2024-05-23

**Rating:** 3
**Confidence:** 3

**Summary:**

This paper presents an MSFNet for the speaker extraction problem using multi-scale feature learning and incorporating the additional EEG signals. Overall, this paper is well organized.

**Strengths:**

1. A multi-scale fusion network for brain-controlled speaker extraction task is proposed.
2. Experimental results demonstrate that the proposed MSFNet model achieves significant improvements over the baseline methods

**Limitations:**

1. In the 'Contributions' of the introduction Section, the authors mentioned that one of the main contributions is to conduct a new AVED dataset. It is essential to clarify whether this dataset is publicly accessible. If the dataset cannot be made available, it may not be appropriate to claim it as a contribution. However, the authors could emphasize their contribution as the construction scheme of the dataset, which can still be valuable to the community by providing a blueprint for others to follow in creating similar datasets.
2. The comparative experiments presented in the manuscript are not sufficiently comprehensive, as the authors have only compared a few EEG-integrated speech enhancement methods. It is noted that these methods are primarily focused on the task of speech enhancement. Are the 'speech enhancement' tasks addressed in these methods equivalent to the 'speaker extraction' task that is the focus of the current work?
3. To provide a thorough evaluation and to substantiate the novelty and effectiveness of the proposed approach, it is recommended that the authors expand the comparisons with traditional speaker extraction approaches, e.g., cascaded speech separation and the speaker verification system.
4. The ablation study presented in the manuscript is not sufficiently robust to conclusively demonstrate the significance of EEG signals within the framework. To rigorously validate the contribution of EEG signals to the overall performance, it is recommended that the authors conduct an ablation study where the EEG Encoder component is removed from the system.

**Suitability:**

2

---

### Official Review · Reviewer_EUB1 · 2024-06-03

**Rating:** 3
**Confidence:** 4

**Summary:**

authors proposed Multi-Scale Fusion Network for Brain-Controlled Speaker Extraction by utilizing speech and EEG signal data.

**Strengths:**

Authors proposed Audio-Video EEG dataset for research purposes. Authors also presented a framework to extract speaker infromation form the proposed dataset

**Limitations:**

MSFNet: Multi-Scale Fusion Network for Brain-Controlled
Speaker Extraction

Authors proposed multiscale fusion network for speaker extraction. However, the Figure 1 does not include any fusion network that authors are proposing in the paper.

This paper mainly focuses on multi-scale fusion network, authors should give details about CMCA-based fused. Authors should give refence to CMCA approach.


In CMCA, the left and right branches handle the audio and EEG streams. Not sure what are these left and right branches authors are talking about here?

What is SI-SDR loss function? Author should give the abbreviation.

we divide all trials of each subject into training, validation, and test sets with proportions of 75%, 12.5%, and 12.5%, respectively. Usually train and test data should be 80-20 or 90-10 split and authors can divide train dataset into train and validation purposes. Is there any specific reason why authors split the dataset where 12.5% is used for testing.

Authors should give more details about how many speakers are used for training, validation and testing purposes.

What is UBESD in EEG signal processing section. Author are using acronyms throughout the paper without defining and not citing the reference.

Figure 3. is this figure plotted for test data alone?

For speech encoder authors are using different window lengths? Is this window applied during STFT? Why authors are referring to kernel size? What is the motivation for using multiple window sizes?

Table 2  in the caption authors should provide acronyms for metrics and an arrow that indicates lower or higher is the better for speaker extraction model.

Ablation study should include with and without speaker extraction module, with and without EEG encoder  with CMCA fusion
 I think the experimental setup should include a systematic way to evaluate the proposed model for speaker extraction.

Also authors should give extracted samples maybe in a GitHub repository for evaluation purposes.

**Suitability:**

2

---

### Meta-Review · Area_Chair_7Zer · 2024-07-01

**Recommendation:** Accept (Poster)
**Confidence:** 4

**Metareview:**

This paper presents an MSFNet for the speaker extraction problem using multi-scale feature learning and incorporating the additional EEG signals. Three out of the four Reviewers tend toward accepting the paper  finding the proposed idea and solution of interest.  Based on this, I suggest accepting the paper as a poster.